Boll characteristics and yield of cotton in relation to the canopy microclimate under varying plant densities in an arid area

Zhang Na 1
Tian Liwen 2
Feng Lu 3
Xu Wenxiu xjxwx@sina.com 1
Li Yabing criliyabing@163.com 3
Xing Fangfang 3
Fan Zhengyi 3
Xiong Shiwu 3
Tang Jianghua 1
Li Chunmei 1
Li Ling 1
Ma Yunzhen 1
Wang Fang 1
1 College of Agronomy/Engineering Research Centre of Cotton Ministry of Education, Xinjiang Agricultural University , Urumqi , Xinjiang , China
2 Cash Crop Research Institute, Xinjiang Agricultural Academy , Urumqi , Xinjiang , China
3 State Key Laboratory of Cotton Biology, Institute of Cotton Research of Chinese Academy of Agricultural Sciences , Anyang , Henan , China
Atif Rana Muhammad
Electronic publication date: 2021 Dec 2
Publication date: 2021
Volume: 9
Electronic Location ID: e12111
Received 2021 May 28; Accepted 2021 Aug 14
Copyright: ©2021 Zhang et al.
Copyright year: 2021
Copyright holder: Zhang et al.
License: This is an open access article distributed under the terms of the Creative Commons Attribution License, which permits unrestricted use, distribution, reproduction and adaptation in any medium and for any purpose provided that it is properly attributed. For attribution, the original author(s), title, publication source (PeerJ) and either DOI or URL of the article must be cited.
License URL: https://creativecommons.org/licenses/by/4.0/

Keywords: Yield-density relationship, Boll distribution, Plant density, Fraction of light intercepted, Canopy temperature and humidity

Funding: The National Key Research and Development Program of China No. 2020YFD1001001 This work was supported by the National Key Research and Development Program of China (No. 2020YFD1001001). The funders had no role in study design, data collection and analysis, decision to publish, or preparation of the manuscript.

==============================
Planting density affects crop microclimate and intra-plant competition, playing an important role on yield formation and resource use, especially in areas where the cotton is grown at relatively high plant densities in Xinjiang, China. However, more studies are needed to examine how the change in planting density affects the microclimate factors such as the fraction of light intercepted (FLI), air temperature(T) and relative humidity (RH) within different canopy layers, which in turn affect the boll number per plant (BNF), boll number per unit area (BNA), boll weight (BW), and boll-setting rate (BSR) at fruiting branch (FB) positions FB1–3, FB4–6, and FB≥7 in cotton. To quantify the relationships between boll characteristics, yield, and microclimate factors, we conducted a 2-year field experiment in 2019–2020 in Xinjiang with six plant densities: 9 (P1), 12 (P2), 15 (P3), 18 (P4), 21 (P5), and 24 (P6) plants m−2. With each three plants m−2 increase in density, the average FLI and RH across different canopy layers increased by 0.37 and 2.04%, respectively, whereas T decreased by 0.64 °C. The BNF at FB≥ 7, FB4–6, and FB1–3 decreased by 0.82, 0.33, and 0.5, respectively. The highest BNA was observed in the upper and middle layers in the P4 treatment and in the lowest canopy layer with the P5. The highest BW was measured in the middle canopy layer for P3, and the highest BSR was measured in the lower layer for P3. Plant density exhibited linear or quadratic relationships with FLI, T, and RH. Microclimate factors mainly affected the boll number in each layer, but had no significant effects on the BW in any layer or the BSR in the middle and lower layers. Cotton yield was non-linearly related to plant density. The 2-year maximum yield was achieved at a plant density of 21 plants m−2, but the yield increase compared to the yield with a density of 18 plants m−2was only 0.28%. Thus, we suggest that the optimal plant density for drip-irrigated cotton in Xinjiang is 18 plants m−2, which could help farmers grow machine-harvested cotton.

Introduction

Cotton (Gossypium hirsutum L.) is an important cash crop grown worldwide as a major source of fibre (Constable & Bange, 2015). China is one of the largest producers and consumers of cotton globally (Mao & Li, 2016). China’s cotton imports, total supply, and use were higher than those of other cotton-producing nations including Brazil, India, and Pakistan (USDA, 2020). Xinjiang Uyghur Autonomous Region has become the most important cotton-growing region in China (Appiah et al., 2014; Tian et al., 2016). In 2020, the region produced 5.2 million tons of lint cotton from 2.5 million planted hectares (NBS, 2020), accounting for 87.33% of the production and 78.93% of the area planted in China. The average lint yield was 2063 kg ha−1, benefiting from intensive management and new cotton varieties (Dai & Dong, 2014; Feng et al., 2017). In Xinjiang, cotton is grown at relatively high plant densities. While increasing plant density increases the cotton yield, it also increases intra-plant competition, resulting in increased shedding and rotten bolls (Bednarz, Nichols & Brown, 2006; Bai et al., 2017). Considering yield and fibre quality for machine-harvested cotton, the cotton planting density must promote “easy, simplified, efficient, and sustainable” production (Dong et al., 2018). However, the optimal machine-harvested plant density under drip irrigation is not clear.

Yield is the combined result of genetic factors and the external environment, whereas microenvironment variation within the canopy affects the ability of the crop to use available resources (Yang et al., 2014). Cotton yield and quality are more susceptible to microclimate conditions than other crops because the reproductive organs are distributed throughout the cotton canopy (Schurr, Walter & Rascher, 2006). Plant density has a strong effect on cotton yield components (Bednarz et al., 2005; Darawsheh et al., 2009), canopy structure (Zhang et al., 2004; Dong et al., 2010; Kaggwa Asiimwe, Andrade Sanchez & Wang, 2013; Chapepa, Mudada & Mapuranga, 2020), light distribution, light interception, air temperature, and humidity within the canopy (Brodrick et al., 2013; Yang et al., 2014; Yao et al., 2017; Xue et al., 2017a; Xue et al., 2017b). Light interception plays a key role in photosynthesis, which is enhanced by a greater photon flux density within the canopy (Aikman, 1989). Light interception is always positively related to dry matter accumulation (Ajayakumar et al., 2017). High cotton planting densities decrease the light distribution in the lower canopy (Brodrick et al., 2013). A moderate planting density (3.0 plants m−2) in the Yangtze River region, which has a mean daily air temperature of 27.1 °C and daily relative humidity of 79.7% from June to October, resulted in high cotton yields (Yang et al., 2014).

Agriculture has strong regional characteristics. The cotton planting density in Xinjiang ranges from 15 to 30 plants m−2 (Dong et al., 2018), which is much higher than in other cotton-producing regions in China. Different climatic conditions, planting densities, and management measures will inevitably result in different growth microclimates. Especially with intensifying climate warming, crop growth and yield are significantly affected (Cammarano and Tian, 2018; Fahad et al., 2021b). As part of the arid zone in central Asia, Xinjiang is extremely scarce of water resources and sensitive to global climate change (Yao et al., 2018). Since 1997, the climate in Xinjiang has shifted from warm and wet to warm and dry (Yao et al., 2021). Given the intensive management of high planting density cotton in Xinjiang’s extremely arid climate, it is necessary to study how altering the planting density affects the cotton canopy microclimate and boll setting characteristics.

Cotton bolls located at different fruiting branch (FB) positions experience different climate conditions (Liu et al., 2015) and boll weight and fibre quality differ at different FB positions (Zhao et al., 2011; Zhao & Oosterhuis, 2000). To assess the optimal plant density under drip irrigation with machine harvesting, we conducted a 2-year field experiment with planting densities of 9 to 24 plants m−2. Our objectives were to clarify the relationships among planting density, canopy microclimate, and yield under extremely arid conditions in Xinjiang, and to determine optimal planting density for machine-harvested drip-irrigated cotton in Xinjiang.

Materials and Methods

Experimental site

The 2-year field experiment was conducted in 2019 to 2020 at the experimental station of the Institute of Cotton Research of the Chinese Academy of Agricultural Sciences in Alaer, Xinjiang (40°60′N, 81°31′E, altitude 1100 m.a.s.l.). The mean annual air temperature at the experimental site ranges from 8.4 °C to 11.4 °C, and the annual accumulated temperature of ≥ 10 °C ranges from 3450 °C to 4432 °C (Li, 2016, Jabran & Chauhan, 2020). The frost-free period lasts 180 to 221 days, and the mean annual precipitation is 48 mm. The monthly precipitation and mean temperature during the 2019 and 2020 cotton growing seasons are shown in Table 1. The soil is sandy loam, and the soil nutrient concentrations at a depth of 20 cm prior to sowing are listed in Table 2.

Experimental design and field management

The experiment was established using a randomized complete block design with three replicates of each planting density of 9 (P1), 12 (P2), 15 (P3), 18 (P4), 21 (P5), and 24 (P6) plants m−2. The plant distances for the six densities were 29.2, 21.9, 17.5, 14.6, 12.5, and 10.9 cm, respectively. The crop row orientation was north–south. Row spacing was wide+narrow i.e., 66 cm+10 cm, and the rows were covered with a 2.05 m wide transparent plastic film. Each plot was 47.9 m2 (7 × 6.84 m). The edges of the film were buried in the soil, leaving a 0.23 m wide bare soil between each sheet. The planting pattern, drip irrigation layout and film cover are illustrated in Fig. 1.

Table 1 Meteorological conditions during the cotton growing seasons in 2019 and 2020.

Variable	Year	April	May	June	July	August	September	October	
Precipitation (mm)	2019	4.70	16.70	28.70	3.20	13.60	26.10	0.00	
	2020	0.20	0.00	7.40	13.20	4.80	1.60	–	
Mean temperature (°C)	2019	19.00	19.80	22.60	26.80	24.90	19.50	12.20	
	2020	17.07	20.78	22.80	23.30	23.60	19.10	–	

Table 2 Nutrient contents of the experimental plot soil in 2019 and 2020.

Year	Total nitrogen (g kg−1)	Organic matter (g kg−1)	Available nitrogen (mg kg−1)	Available phosphorous (mg kg−1)	Available potassium (mg kg−1)	
2019	0.40	9.98	21.00	32.01	72.00	
2020	0.48	10.02	51.40	36.70	94.00	

Figure 1 The planting pattern and drip irrigation pipe layout.

The cultivar used in the experiment was hybrid cotton variety CRI88 with a growth duration of approximately 136 days. Cotton was sown on 18 April 2019 and 21 April 2020 using the manual hill-drop method after covering the rows with plastic film. Seedlings were manually thinned at the two-leaf stage to obtain the desired planting densities. The buds of the main stem were topped on 17 July 2019 and 13 July 2020. The cotton was harvested on 15 October 2019 and 03 October 2020. Before sowing, fertilizer was applied at 4.8 t ha−1 organic fertilizer, 225 kg ha−1 urea (46.4% N), and 300 kg ha−1 primary calcium phosphate (46% P2O5). Fertilizer consisting of 150 kg ha−1 urea, 270 kg ha−1 diammonium phosphate (18% N, 46% P2O5), and 112.5 kg ha−1 potassium dihydrogen phosphate (52% P2O5, 34% K2O) was applied as a top dressing with each irrigation. The plots were irrigated nine times over the growing period with a total of 4200 m3 ha−1. Other management actions followed the local farming practices.

Data collection

Fraction of light intercepted within the canopy

Fraction of light intercepted (FLI) within the canopy was evaluated from the budding to boll opening stage in 2019 and 2020. Incident photosynthetically active radiation (PAR0) and transmitted photosynthetically active radiation (PARc) were measured using a LI-191SA light quantum sensor and a LI-1400 data logger (LI-COR, Lincoln, NE, USA). The canopy was divided into 0.2 m ×0.2 m vertical and horizontal grids. The quantum sensor was placed perpendicular to the rows, and three replicate photosynthetically active radiation measurements were taken in each plot. The intercepted light rate (Ir) of each sensor was computed using Eq. (1). FLI was computed according to the Simpson 3/8 integration rule (Xue et al., 2017a), using Eqs. (2) and (3), where Ai isthe amount of light in a certain cross-sectional area, the coefficient vector is {1,3,3,2,3,3,2, …,3,3,2,1}, Δx is the vertical interval of the grid, Δy is the horizontal interval, i and j are grid node numbers, and G(i,j) represents kriging interpolation points, FLI is the total light interception rate in the certain area of the canoy. The canopy was divided into lower, middle, and upper layers as shown in Fig. 2. (1) Ir=1−PARc/PAR0

(2) Ai=3Δx8Gi,1+3Gi,2+3Gi,3+2Gi,4+...+2Gi,ncol−1+Gi,ncol

(3) FLI≈3Δy8A1+3A2+3A3+2A4+...+2Ancol−1+Ancol

Figure 2 Vertical distribution of cotton canopy layers.

Canopy air temperature and relative humidity

Canopy air temperatuer (T) and relative humidity (RH) were monitored with an automatic Lascar EL-USB-2 data logger (Lascar Electronics, Erie, PA, USA). The sensors were installed at approximately 1/3, 1/2, and 2/3 of the canopy height at the position between wide and narrow rows at full squaring, and at the second FB (FB2), fifth FB (FB5), and eighth FB (FB8) after the full blooming period. The data recorded every 30 min from 10:00 to 21:00 and averaged to daily mean values.

Spatial boll distribution

On 10 October 2019 and 28 Septemper 2020, 30 plants in each plot were selected to determine the spatial boll distribution. Bolls were divided into three groups according to whether they were found on FBs 1–3 (FB1−3), FBs 4–6 (FB4−6), and FBs higher than 7 (FB≥7). Bolls number per plant were collected from FB1−3, FB4−6, and FB≥7 in each plot. The boll-setting rate (BSR) for different FBs was equal to number of setting bollsdivided by the total number of fruit nodes. Individual boll weights (BWs) at different FBs were determined after drying the bolls in the sun to a constant weight.

Seed cotton yield

Seed cotton in the area of 14.35 m2 (7 ×2.05 m) with three repetitions were handpicked on 15 October 2019 and 3 October 2020, and weighed after sun-drying.

Data analyses

SPSS 25.0 software (SPSS Inc., Chicago, IL, USA) was used to run non-linear regression and ANOVA. The least significant difference (LSD) test at the 0.05 level was used to compare the mean of different treatments. Graphics were created using origin 2018 graphics software (Origin LabInc., Northampton, MASS, USA).

Results

FLI within the canopy

FLI within the canopy increased with the planting density, but decreased with the increase in canopy height (Fig. 3). Over the entire growth period, the maximum FLI in the upper layer was observed in the P5 (0.66) in the full-boll period in 2019 and in the P6 (0.35) in the full blooming period in 2020. P5 produced the highest 2-year average FLI in the middle (0.85) and lower layers (0.97) in the full-blooming period. Compared with the peak value in each treatment, FLI was reduced by 0.25–0.39, 0.17–0.40, and 0.07–0.30 in the upper, middle, and lower canopies, respectively, at the boll-opening period. Among the different planting densities, P1 and P2 resulted in the greatest FLI reduction in the upper layer, whereas the smallest FLI reduction was in P4 and occurred in the middle and lower layers.

Figure 3 Distribution of canopy FLI within the canopy in response to plant density.

Distribution of air T within the canopy

Consistent with changes in the outside air T (control [CK]), the air T within the canopy increased and then decreased over the course of the growing season in both years (Fig. 4). For all treatments, T was higher than CK in the upper canopy layer. The higher the planting density, the lower the T within the canopy. Increasing the planting density not only advanced the time when the cooling effect appeared but also increased the cooling rate. At the middle canopy layer, the T of P6 at the full-blooming stage was 0.31 °C lower than CK, while that for P5 was 0.16 °C lower than CK at the full-boll stage. In the lower layer, Ts of P4, P5, and P6 at the full-blooming stage were 1.68, 1.64, and 2.11 °C lower than CK, respectively, while T at P3 was 0.87 °C lower than CK at the full-boll stage.

Figure 4 Distribution of air T within the canopy in response to plant density.

T was higher in the upper canopy layer than in the middle and lower layers, but the depression in T was greater between the upper and middle layers than that between the middle and lower layers. Over the 2 years, the T in the middle canopy layer in the P1–P6 treatments was 4.21, 4.11, 3.19, 3.06, 2.72, and 2.49 °C lower than that in the upper layer, respectively, but 1.29, 1.15, 1.62, 1.49, 1.32, and 1.23 °C higher than that in the lower layer.

Distribution of RH within the canopy

Across canopy layers, RH was highest during the full-boll period (Fig. 5). The peak RH in the upper, middle, and lower layers was 51.66, 63.88, and 70.57% in 2019, respectively, and 52.83, 64.69, and 71.84% in 2020. At boll opening, the respective RH values decreased by 33.85, 36.10, and 37.92% in 2019, and 33.31, 39.65, and 41.84% in 2020 when compared to peak values.

Figure 5 Distribution of RH within the canopy in response to plant density.

Contrary to the variation in T within the canopy, RH throughout the canopy increased with planting density. In the upper layer, the canopy RH was higher than CK in the P5 and P6 plots, whereas it was lower than CK in the P1 plot depending on the growth period. In the middle and lower layers, the canopy RH of all treatments was higher than CK. As the planting density increased, the amplitude of RH variation between the middle and upper layers decreased. The 2-year average RH depression over the entire growth period was 10.16, 10.22, 9.39, 9.02, 8.14, and 8.45% for plots P1 to P6, respectively. The amplitude of RH variation between the middle and lower layers showed no particular trend.

Boll density, single boll weight, and boll setting rate at different FB positions

Increasing the density reduced the number of bolls at different FB positions (Table 3). With each 3-plants m−2 increment, the mean boll number per plant (BNF) at FB≥7, FB4−6, and FB1−3 decreased by 0.83, 0.33, and 0.5 in 2019 and 0.86, 0.55, and 0.38 in 2020, respectively. BNF in plots P1 and P2 differed significantly from BNF in the P5 and P6 plots (P < 0.05) at different FB positions. At FB≥7, the maximum boll number per area (BNA) was greatest in P4 plots in 2019 and P3 plots in 2020, and these maxima were significantly higher than those in the P5 and P6 plots (P < 0.05). At FB4−6, the BNA in P4 plots was significantly higher than those in P1 and P2 plots (P < 0.05), with maxima of 63.0 bolls m−2 in 2019 and 64.8 bolls m−2 in 2020. At FB1−3, the 2-year average BNA was highest in P5 plots (73.19 bolls m−2), and it was also significantly higher than the BNA in P1 and P2 plots (P < 0.05) but not significantly different from the BNA in P6 plots. Boll-setting rates (BSR) declined in the order FB1−3 >FB4−6 >FB≥7. With values of 76.48% at FB1−3 and 59.89% at FB4−6, the 2-year average BSR in P3 plots was significantly higher than those in the other treatments (P < 0.05).

Relationships of planting density to FLI, T, and RH

Under different planting densities, FLI in the middle canopy layer and T and RH in all canopy layers showed linear relationships with planting density. The relationship between FLI in the upper and lower layers and density followed a quadratic curve pattern (Fig. 6). Regression fits are shown in Table 4. Increasing the density had no significant effect on FLI in the upper canopy layer. There was a positive linear relationship between density and FLI in the middle layer, and a significant, negatively correlated conic relationship with FLI in the lower layer. T in each canopy layer declined with increased planting density, whereas RH increased.

Figure 6 Fits of plant density with canopy FLI, T and RH.

Correlations among canopy FLI, T, RH, BNF, BNA, BW, and BSR

As shown in Fig. 7, canopy T and RH in each layer were negatively correlated. In the upper layer, FLI was uncorrelated with T, RH, BNF, BNA, BW, and BSR. T was positively correlated with BNF, BNA, and BSR, whereas RH was negatively correlated with BNF and BSR. In the middle and lower layers, FLI was negatively correlated with T and BNF but positively correlated with RH and BNA. T was positively correlated with BNF but negatively correlated with BNA. RH was negatively correlated with BNF and positively correlated with BNA only in the lower canopy layer.

Figure 7 Correlations of canopy FLI, T, and RH with BNF, BNA, BW, and BSR.

Significant interactions between the boll number, BW, and BSR were mainly found for the upper canopy layer. Among them, BSR was positively correlated with BNF, BNA, and BW, BNF was positively correlated with BNA, and BNA was positively correlated with BW.

Yield

Yield varied greatly with planting density (Fig. 8). The average 2-year yield increased by 0.28–24.33% when the planting density increased from 9 plants m−2 (P1) to 21 plants m−2 (P5). The highest yields were seen for P5 of 6644.52 kg ha−1 in 2019 and P4 of 6517.26 kg ha−1 in 2020. There was no significant difference in the yields of P4 and P5 (P > 0.05), but they were significantly higher than the yields obtained in P1, P2, P3, and P6 in both years (P < 0.05). The relationship between yield and planting density is shown in Fig. 9. The fitting curve was parabolic and opened downwards, and the fitting coefficients R2 were all higher than 0.9 (P < 0.01). The curve simulation also showed that the P4 (18 plants m−2) treatment had the maximum yield.

Figure 8 Seed cotton yield per unit area in 2019 and 2020.

The different small latter above the columnar represents significant differences at P < 0.05.

Figure 9 Cotton yield in response to plant density in 2019 and 2020.

Symbols in each year represent a single harvest seed cotton yield (n = 3).

Discussion

Higher plant density utilized solar radiation, nutrients and space, which ultimately improved the seed cotton yield. Although there were differences in the 2 years, the yield first increased with the plant density and then decreased (Fig. 9). It was highest at planting densities of 18 or 21 plants m−2, but the difference between the two treatments was not significant (Fig. 8). This substantiates the common opinion that increasing the planting density will not make the yield continue to increase. The yield remained approximately the same or even decreased after a certain threshold was reached. The boll distribution at lower plant density increased the bolls at FB≥7, mainly because low-density treatment produced more bolls per plant with more FBs and stem nodes, which enabled more source and sink connections. Boll number per m2 (except FB≥7), BW, and BSR at different FBS were all highest at 15 or 18 plants m−2 (Table 3). These three indicators decreased to varying degrees when the planting density exceeded 18 plants m−2. The yield and boll characteristics were not as good under the crowded conditions encountered at high densities. This may be caused by intensified competition for limited resources and the impoverished environment (Li et al., 2020).

More light was intercepted at higher planting densities in different canopy layers. This confirms the view that high planting densities can help with achieving high levels of radiation interception by the crop (Mao et al., 2014; Zhang et al., 2014). The vertical distribution of light within the canopy was not uniform; it was highest in the lower canopy layer and lowest in the upper layers (Fig. 3). A dense canopy with a high leaf area index was the main reason for the difference (Xue et al., 2017a; Xue et al., 2017b). Moreover, we also showed that the highest light interception rate does not result in the highest yield. Due to the seed cotton yield was relation to light penetration and ventilation into the lower levels of the canopy (Kaggwa Asiimwe, Andrade Sanchez & Wang, 2013; Zhi et al., 2014). An appropriate density allows greater light penetration and gaseous exchange (Meredith, 1984), which improves the utilization of light resources and maintains high crop productivity, and assists in the development of bolls in the canopy (Chapepa, Mudada & Mapuranga, 2020).

Table 3 Effects of planting density on boll number and boll weight at different fruiting branch position in 2019 and 2020.

		Boll number	Boll weight	Boll setting ratio	
Fruiting branch (FB)	Treatment	(per plant)	(per m2)	(g/boll)	(%)	
		2019	2020	2019	2020	2019	2020	2019	2020	
FB1-3	P1	5.03a	5.17a	45.27c	46.53c	6.13c	6.07c	62.92ab	70.14c	
	P2	4.93a	4.93a	59.16b	59.16b	6.34bc	6.48b	64.91a	78.72b	
	P3	3.93b	5.03a	58.95b	75.45a	6.87a	6.94a	68.60a	84.36a	
	P4	3.23c	4.40b	58.14b	79.20a	6.64ab	6.63b	61.78b	80.00ab	
	P5	3.17c	3.80c	66.57a	79.80a	6.07c	6.16c	67.38a	69.09c	
	P6	2.53d	3.27d	60.72ab	78.48a	6.16c	6.34bc	64.96a	58.33d	
FB4-6	P1	4.07b	5.23a	36.63d	47.07c	7.00a	6.92ab	41.78d	58.80a	
	P2	4.67a	4.50b	56.04ab	54.00c	7.06a	7.06ab	56.91ab	61.64a	
	P3	3.50c	3.73c	52.50b	55.95ab	7.07a	7.28a	57.30a	62.57a	
	P4	3.50c	3.60c	63.00a	64.80a	7.17a	7.23a	52.50b	62.43a	
	P5	2.53d	3.00d	53.13b	63.00a	7.04a	6.73b	46.06c	51.14b	
	P6	2.40d	2.47d	57.60ab	59.28ab	6.96a	6.64b	40.45d	42.53c	
FB≥ 7	P1	5.40a	4.90a	48.60ab	44.10b	6.50a	6.33ab	36.65a	52.88ab	
	P2	3.73b	4.83a	44.76bc	57.96a	6.60a	6.50ab	33.04b	49.83b	
	P3	2.70c	3.87ab	40.50c	58.05a	6.63a	6.70a	34.76b	55.50a	
	P4	2.90c	2.57bc	52.20a	46.26b	6.85a	6.53ab	37.83a	43.75c	
	P5	1.23d	1.80cd	25.83d	37.80c	6.55a	6.04bc	25.69c	32.14d	
	P6	1.27d	0.60d	30.48d	14.40d	6.07a	5.54c	26.03c	18.56e	

Plant density alters the characteristics of the boundary between the leaves and surrounding air, as well as affecting canopy T and RH. Higher yields were obtained at a lower canopy T (Han et al., 2007; Fan et al., 2007), and canopy T and yield were negatively correlated in wheat (Amani, Fischer & Reynolds, 1996). In our study, there was negatively correlation between T and plant density, while the RH and plant density was positively correlated (Table 4), which means that T in the different canopy layers decreased with increasing planting density (Fig. 4), whereas RH increased (Fig. 5). This may be due to the fact the ground in high plant density is shaded from the sun, and the evapotranspiration is higher in higher plant density, resulting in lower canopy T. While lower plant density with an open canopy could allow more air flow for evaporation, thereby reducing RH. Therefore, increasing plant density played a significant role in cooling and humidifying the canopy environment.

Table 4 Regression equations of canopy variables with plant density.

Layer	Factor	Fitting equation	Correlation coefficient	P-value	RMSE	
Upper	Fraction of light intercepted (%)	y = −0.0456x2+ 1.6132x + 13.792	0.8602	0.05	0.66	
	Temperature (°C)	y = −0.2885x + 38.867	0.9951	0.00	0.13	
	Relative humidity (%)	y = 0.7742x + 30.055	0.9757	0.00	0.77	
Middle	Fraction of light intercepted (%)	y = 0.67x + 44.774	0.7143	0.03	2.66	
	Temperature (°C)	y = −0.1663x + 33.555	0.9851	0.00	0.13	
	Relative humidity (%)	y = 0.6271x + 41.711	0.9690	0.00	0.70	
Lower	Fraction of light intercepted (%)	y = −0.1052x2+ 4.6195x + 31.291	0.9538	0.01	1.97	
	Temperature (°C)	y = −0.1684x + 32.22	0.9744	0.00	0.17	
	Relative humidity(%)	y = 0.6054x + 46.512	0.9671	0.00	0.70	

We also found that compared with BW and BSR, boll number was significantly affected by canopy FLI, T, and RH (Fig. 7). Of these, FLI and RH were negatively related to boll number at the single-plant level, but positively related to population boll number, while the relationship between T and boll number was opposite that of RH. This means that the canopy microclimate of lower T, higher RH, and higher FLI formed at higher planting densities, especially in the middle and lower canopy layers, was not conducive to boll retention. This also explained why BW and BSR in the middle and lower canopy layers of the 18 and 21 plants m−2 treatments were lower than in the other treatments.

Conclusions

Increasing the planting density resulted in increased FLI and RH and a decrease in T in different canopy layers. Microclimate factors mainly affected the boll number in each layer significantly, but had no significant effects on BW in any layer or BSR in the middle and lower layers. The canopy microclimate of lower T, higher RH, and higher FLI formed at high planting densities negatively affected boll number, BW, and BSR. Although the 2-year average seed cotton yield was highest at 21 plants m−2, it was only 0.28% higher than at 18 plants m−2, and the difference was not significant (P > 0.05). Thus, we suggest that when using a 66 cm+10 cm planting pattern with drip irrigation and film mulching in Xinjiang, the appropriate planting density is 18 plants m−2.

Supplemental Information

Supplemental Information 1 Raw data

Click here for additional data file.

We thank the reviewers Shah Fahad and Muahammad Tehseen Azhar, professors Lizhen Zhang, Honghai Luo and Penghao Wu for their constructive comments on a previous version of this manuscript.

Additional Information and Declarations

Competing Interests

Author Contributions

Data Availability

The authors declare there are no competing interests.

Na Zhang performed the experiments, analyzed the data, prepared figures and/or tables, authored or reviewed drafts of the paper, and approved the final draft.

Liwen Tian performed the experiments, analyzed the data, authored or reviewed drafts of the paper, and approved the final draft.

Lu Feng conceived and designed the experiments, performed the experiments, authored or reviewed drafts of the paper, and approved the final draft.

Wenxiu Xu and Yabing Li conceived and designed the experiments, authored or reviewed drafts of the paper, and approved the final draft.

Fangfang Xing, Zhengyi Fan, Shiwu Xiong, Jianghua Tang, Chunmei Li, Ling Li, Yunzhen Ma and Fang Wang performed the experiments, authored or reviewed drafts of the paper, and approved the final draft.

The following information was supplied regarding data availability:

The raw data are available in the Supplemental File.

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
