# Peer review of "Boll characteristics and yield of cotton in relation to the canopy microclimate under varying plant densities in an arid area"

_PeerJ, doi:10.7717/peerj.12111_

## Round 0.1 · original submission · Major Revisions

Please address the Reviewer's comments. The language of the manuscripts needs editing. The Results, as well as Discussion portions, should be elaborated.

·

Basic reporting

The manuscript need English editing.

Experimental design

OK

Validity of the findings

OK

Additional comments

English should improve by a native person. The paper suffers from a poor English structure throughout and cannot be published or reviewed properly in the current format. The manuscript requires a thorough proofread by a native person whose first language is English. The instances of the problem are numerous and this reviewer cannot individually mention them. It is the responsibility of the author(s) to present their work in an acceptable format. Unless the paper is in a reasonable format, it should not have been submitted.
2. The novelty of the study needs to be highlighted compare to other similar studies.
3. Discussion is weak. The discussion needs enhancement with real explanations not only agreements and disagreements. Authors should improve it by the demonstration of biochemical/physiological causes of obtained results. Instead of just justifying results, results should be interpreted, explained to appropriately elaborate inferences. Discussion seems to be poor, didn't give good explanations of the results obtained. I think that it must be really improved. Where possible please discuss potential mechanisms behind your observations. You should also expand the links with prior publications in the area, but try to be careful to not over-reach. For the latter, you should highlight potential areas of future study.
4. The scientific background of the topic is poor. In "Introduction" and "Discussion", the authors should cite recent references between 2016-2020 from JCR journals.
Fahad, S., Sönmez, O., Saud, S., Wang, D., Wu, C., Adnan, M., Turan, V. (Eds.), 2021a. Plant growth regulators for climate-smart agriculture, First edition. ed, Footprints of climate variability on plant diversity. CRC Press, Boca Raton, FL.
Fahad, S., Sonmez, O., Saud, S., Wang, D., Wu, C., Adnan, M., Turan, V. (Eds.), 2021b. Climate change and plants: biodiversity, growth and interactions, First edition. ed, Footprints of climate variability on plant diversity. CRC Press, Boca Raton.
Fahad, S., Sonmez, O., Saud, S., Wang, D., Wu, C., Adnan, M., Turan, V. (Eds.), 2021c. Developing climate resilient crops: improving global food security and safety, First edition. ed, Footprints of climate variability on plant diversity. CRC Press, Boca Raton.
Fahad, S., Sönmez, O., Turan, V., Adnan, M., Saud, S., Wu, C., Wang, D. (Eds.), 2021d. Sustainable soil and land management and climate change, First edition. ed, Footprints of climate variability on plant diversity. CRC Press, Boca Raton

·

Basic reporting

I must congratulate to the authors of this manuscript for conducting this study, I mean such study are very limited in the literature. I have gone through this article, when I just the file sentence of "ABSTRACT" portion, that confused me, where it is mentioned "making full use of the group effect are the keys to achieving high-efficiency ". This is not clear to anyone, this indicate the re-phrase of such sentence to remove the ambiguity (if any). It allowed me for mentioning that, this article must ne edited for English Native Speaker before publication.
Likewise, the author should elaborate the RESULTS and DISCUSSION portion, and this is possible after see critical review of tables, figure and statistical analysis.

Experimental design

Well designed

Validity of the findings

I have also gone through the data and its statistical analysis, this portion is acceptable to me

---

## Round 0.2 · accepted · Accept

All the queries raised by the Reviewers have been addressed, thus the manuscript is being accepted for publication.

·

Basic reporting

ok

Experimental design

ok

Validity of the findings

ok

Additional comments

Accepted as it is